# Genetic clustering within massive *Porites* species complex is the primary driver of holobiont assembly

Carly B. Scott[1]*, Raegen Schott[2], Mikhail V. Matz[2]

**1** Department of Biology, University of North Carolina at Chapel Hill, Chapel Hill, North Carolina, United States of America, **2** Department of Integrative Biology, University of Texas at Austin, Austin, Texas, United States of America

* cabscott@unc.edu

## Abstract

The fate of coral reefs in response to climate change depends on their ability to adapt to new environments. The coral animal is buffered from environmental stress by its algal endosymbionts and microbial partners (together, the "holobiont"). However, the flexibility of holobiont community assembly is not well understood, making it difficult to estimate its contribution to coral adaptation. To clarify these processes, we genetically profiled holobiont components (coral, algal symbiont, and microbiome) of massive *Porites* sampled across two size classes (small, < 30 cm and large, > 2 m) and ecologically distinct reef sites near Orpheus and Pelorus Islands, Australia. We recovered five major genetic clusters in the coral host. We estimated the relative contributions of the host genetic structure, site, and size class to holobiont community composition. Host genetic structure was the primary driver of both Symbiodiniaceae and microbial communities, indicating strong holobiont specificity in genetic clusters. In addition, the microbial community was associated with reef site and size class, unlike Symbiodiniaceae that were not significantly affected by either factor. As environmentally segregated, cryptic genetic lineages emerge as a common feature of scleractinian corals, these results emphasize that failure to assess cryptic genetic structure of the coral host may lead to dramatic overestimation of holobiont flexibility.

## Introduction

The coral holobiont, or the collective unit of the coral animal, its intracellular algal symbionts, and its associated microbial community, determines how coral reefs respond to environmental change [1]. Studies have posited that specific holobiont communities may increase the resilience of the coral to stress [2]. Given the acceleration of global change, acclimatization to future conditions through microbial community shifts has garnered significant attention as a hopeful mechanism for reef

**Data availability statement:** All sequencing (2bRAD and 16S/1TS2 amplicon) data generated are located at NCBI BioProject PRJNA1048506. All scripts and intermediate products (including metadata) used to complete the analysis are found at https://github.com/cb-scott/PoritesHolobiont_Final and archived on Zenodo at doi: 10.5281/zenodo.15565326.

**Funding:** This research was supported by NSF DGE 2137420 to C.B.S, NSF grant IOS-1755277 to M. V. M., funding from the University of Texas Department of Integrative Biology to C.B.S, and funding from the International Women's Fishing Association to C.B.S. The funders had no role in the study design, data collection and analysis, decision to publish, or preparation of the manuscript.

persistence [3–5]. However, the underlying mechanisms which determine coral holobiont assembly are still not fully understood.

Without understanding the drivers of coral-associated communities, it is difficult to assess their acclimatization potential. It is likely that genetic, spatial, and temporal factors interact to drive the coral holobiont [6]. Coral taxa exhibit distinct "core" holobiont members in the same environments, implying the functional importance of certain microbial species to the reef [7,8]. Similarly, species exhibit geographic variability in their microbial and microalgal associations, indicating a degree of local acclimatization [9].

Over-time studies of the holobiont response to stressors (e.g., bleaching) have yielded conflicting results. Stress events generally induce holobiont reassembly for both Symbiodiniaceae and microbial communities, but the degree and permanence of these shifts is highly variable [10–13]. For Symbiodiniaceae, studies have reported both stable switches to a new symbiont association [14,15] and a return to the native state after the stress event [16,17]. The degree of variability of the coral microbiome is also not well established, but the effect of time is often dwarfed by environmental and species-specific pressures [6,18,19].

It has been proposed that coral holobiont dynamics are ungeneralizable and likely geographically and genotypically unique [6,20,21]. While this is possible, most studies fail to integrate individual-level genotyping, size class, and site data to disentangle these effects. Thus, the relative roles of host genotype, environment, and time have yet to be resolved at ecologically relevant scales.

Here, we compared host-associated communities of massive *Porites* across size classes (small, < 30 cm and massive, > 2 m), sites (spanning 13 km), and five genetic clusters. The massive *Porites* species complex forms the structure of many Pacific reefs, growing up to 17 meters in diameter and living over 500 years [22,23]. Additionally, massive *Porites* are broadcast spawning species which transmit their Symbiodiniaceae vertically through eggs and generally transmit their microbial community horizontally [24]. Thus, we expect the Symbiodiniaceae community to be tightly associated with host genetics and the microbial community to be strongly affected by the local environment. We sampled small (<30 cm) and large (>2 m and up to 10 m) massive *Porites* colonies at four sites across Orpheus and Pelorus Islands, Australia. Despite their close proximity, these sites exhibit qualitatively different environments: Northeast and Southeast Pelorus (NE and SE Pelorus, respectively) are deeper sites with generally high visibility and high wave action, Pioneer Bay (PB) is a shallow, nearshore, and highly turbid site, and South Orpheus (S Orpheus) is also a shallow nearshore site but with fast currents due to its channel location. From this we aimed to, (i) investigate the genetic structure of the coral hosts, and (ii) estimate the relative importance of this genetic structure, site, and size class for the composition of the associated Symbiodiniaceae and microbial communities.

## Materials and methods

### Sampling

*Porites* spp. were sampled around Orpheus and Pelorus Islands, Australia in November 2019, under GBRMPA permit G18/41245.1. To estimate the effects of size on

holobiont structure, we sampled massive adults (>2 m diameter) and their smaller counterparts (<30 cm diameter; Fig 1). Three samples were taken across each massive individual (one central and two on opposing sides), and one sample was taken from each putative juvenile. Samples were collected from four sites around the two islands exhibiting different environments (Fig 1). In total, we collected 193 samples from 90 colonies. Coral fragments were fixed immediately after sampling in 100% EtOH and subsequently stored at -80°C.

## DNA library preparations

DNA was extracted from all samples using the CTAB method (see supplemental methods). 2bRAD sequencing libraries (https://github.com/z0on/2bRAD_denovo) were constructed for each colony and sequenced using Ilumina NextSeq 500 SR 75 at the Genomic Sequencing and Analysis Facility at UT Austin. Amplicon sequencing libraries were constructed for all samples. ITS2 amplification used primers from [26] (SymVarF: CTACACGACGCTCTTCCGATCTGAATTGCAGAACTC-CGTGAACC, SymVarR: CAGACGTGTGCTCTTCCGATCTCGGGTTCWCTTGTYTGACTTCATGC); 16S V3/V4 region was amplified using primers from [27] (16S_341F_Truseq: CTACACGACGCTCTTCCGATCTCCTACGGGNGGCCTACG-GGNGGCWGCAG, 16S_805R_Truseq: CAGACGTGTGCTCTTCCGATCTGACTACHVGGGTATCTAATCC). All amplicon

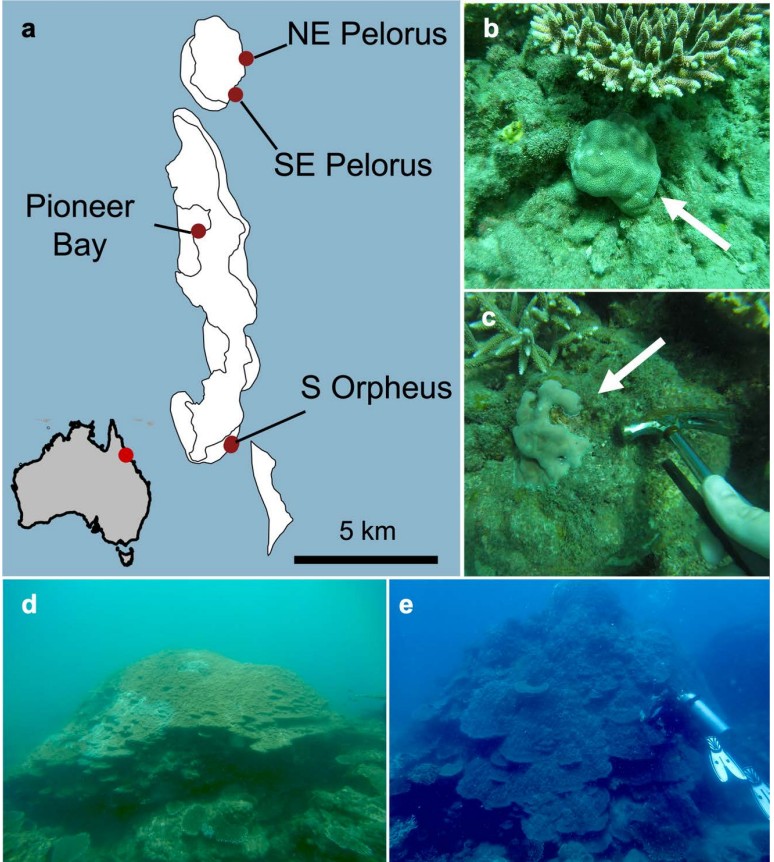

**Fig 1. Sampling location and example *Porites* spp. size classes.** (a) Map of Orpheus and Pelorus Islands, Australia. These islands are located on the inner Great Barrier Reef, and their relative location to Australia is marked on the map inset. (b, c) Examples of colonies we considered small *Porites* spp., less than 30 cm in diameter. (d, e) Individuals we considered large *Porites* spp., greater than 2 m in diameter. Individual colonies were not measured due to the large, visible disparity between what we considered size classes. Orpheus and Pelorus Island map data edited from © Great Barrier Reef Marine Park Authority 2014 [25].

sequencing libraries were amplified with 20–30 PCR cycles, depending on band intensity. Negative controls (MilliQ water) were run with each set of samples and amplification was never observed. Libraries were pooled according to visually assessed gel band brightness. Pooled amplicon libraries were sequenced on the MiSeq PE150.

## Bioinformatic pipeline

Reads were trimmed of adaptors and deduplicated using custom 2bRAD scripts (https://github.com/z0on/2bRAD_denovo). Reads containing base calls with quality less than Q15 were removed using *cutadapt* [28]. Cleaned reads were mapped to a combined reference of the *Porites lutea* genome [29], *Symbiodinium* sp. [30], *Breviolum minutum* [31], *Cladocopium* sp. [30], and *Durusdinium* sp. [32] genomes using *bowtie2* with default local settings [33]. Mapped reads were filtered for a mapping quality >30 and genotyping rate >50% and only reads which mapped to *Porites lutea* were retained. Between-sample genetic distances were calculated as identity-by-state (IBS) based on single-read resampling in ANGSD [34].

   At this point we subset the data to remove technical replicates, colony replicates, and clones based on hierarchical clustering, retaining one individual per genotype (N = 65). After genetically identical samples were removed, the genetic distances were calculated again in ANGSD with the same filtering scheme. We then used 'pcangsd' to calculate the most likely number of admixture clusters in our host samples [35].

## Host genetic analysis

Initial analysis revealed highly distinct genetic/admixture groups within *Porites* spp. We calculated pairwise $F_{ST}$ between each admixture group using the realSFS module of ANGSD [34]. We visualized these pairwise $F_{ST}$ values as an unrooted hierarchical clustering tree, using the UPGMA algorithm in R package 'ape' [36].

   We used RDA/PERMANOVA analysis to determine the relative contribution of site and size class to *Porites* spp. genetic structure. Using the 'adonis2' function from the R package 'vegan' [37] we tested for significance of age and site in predicting genetic distance between individuals using 'method = marginal' (Genetic Distance ~ Site + Size Class + Site*Size Class).

## Symbiont and microbe ASV prefiltering

We used the same basic computational workflow for both 16S and ITS2 sequences. Primers and adapters were trimmed from reads with cutadapt. For microbes, likely host contamination was removed from our samples using METAXA2 [38]. We only retained reads classified as bacterial or archaeal in origin.

## Determination of symbiont ASVs

Symbiodiniaceae ITS2 locus are difficult to deal with due to their intragenomic variation and multicopy nature. While the SymPortal [39] framework has been proposed to handle these challenges, our reads did not have sufficient overlap between the left and right reads of the pair to be retained through SymPortal prefiltering steps. Thus, ITS2 sequence variants were identified via the DADA2 pipeline in R statistical software [40]. To deal with the high degree of intragenomic variation expected in Symbiodiniaceae ITS2 sequences, we calculated the correlation of ASV presence/absence across samples. Under the logic that highly correlated ASV sequences likely correspond to within-species intergenomic variation, rather than additional species diversity, we clustered any ASVs with correlated presences higher than 0.8 into a single "ASV group" which was used in downstream analyses.

   To assign taxonomy to the ITS2 ASVs, we constructed a custom blast database from the SymPortal DIV database (updated on 2024-02-13). We then blasted our ASV sequences against this database, keeping the best 30 hits per ASV. From the blast classifications, we retained only the subtype of each strain's entry in the SymPortal database (e.g., C15 rather than C15au). We filtered the blast results for hits with an e-value < 1e-100 and >95% match to the target sequence. At this threshold, some queries still had multiple best hits. For these queries, we determined if there was consensus in

their best hits at the subtype-level (e.g., all C15, but different strain identifiers). We assigned a query ASV this subtype if there was over 90% consensus in the blast hits with e-value < 1e-100 and >95% match to the target. For ITS2 ASVs not meeting this criterion, we did not assign taxonomy.

To aggregate our colony-level replicates, we calculated the sum of ASV groups across replicates to gain a "colony-averaged" community. ASV groups were retained if they were present in at least three colonies and had > 100 reads across samples. Similarly, colonies were retained if they had > 3 nonzero ASVs present and >1000 total sequenced reads.

### Determination of microbial ASVs

The DADA2 pipeline was also used on trimmed microbial reads for denoising and ASV classification. We assigned taxa to ASVs in DADA2 using the 'assignTaxa' function, using the Silva version 138.1 SSU reference database (DADA2 formatted database taken from: https://zenodo.org/records/4587955). Similar to the ITS2 ASV filtering, we obtained a colony-level community measure by summing across ASVs in colony replicates. We only retained ASVs present in >3 colonies with a minimum of 100 reads across colonies. We only kept colonies with >1000 total sequenced reads and >3 nonzero ASVs present.

### Community-level analysis

The same analytical pipeline was used for both ITS2 and 16S ASVs in parallel. Given we wanted to evaluate the role of host genetics in structuring the holobiont community, we only retained colonies which had corresponding 2bRAD sequencing data. ASV abundances were then standardized across colonies using the Hellinger method using the 'decostand' function from vegan. Bray-Curtis distances between colonies were calculated from Hellinger-standardized abundances using 'vegdist'.

We evaluated the role of host genetic structure, site, and size class in two different ways. First, we conducted a PERMANOVA analysis using vegan's 'adonis2' function, with option "method=marginal". We used the same model for each community distance matrix (Holobiont Community Distance ~ Host Admixture Group + Site + Size Class). In parallel, we conducted an RDA-forest analysis (https://github.com/z0on/RDA-forest) [41]. RDA-forest is an application of gradient forest (a multidimensional extension of random forest) to multi-dimensional data, with the goal of predicting PCoA structure [42]. We first constructed an unconstrained PCoA from community distances using vegan's 'capscale' command. The first fifteen principal coordinates were used as the response matrix, and the predictor matrix included host admixture group, site, and size class. Categorical predictors with multiple levels (site and genetic cluster) were converted into dummy quantitative variables with values 1 and 0, with one variable per predictor level. Each random forest model used 1500 bootstrapped trees and max variable permutation level of $\log2(N_{Individuals} * 0.368/2)$. Then, for multi-level categorical predictors, we summed the weighted importance across dummy variables representing their levels to obtain overall importance.

### 16S Differential abundance analysis

Differential abundance of microbes was calculated using DESeq2 on the raw abundance matrix with the experimental design, Abundance$_{OTU}$ ~ Admixture Group + Site + Age [43]. We designated $\alpha < 0.01$ as the cutoff to determine which taxa were significantly differentially abundant between comparisons.

All results were visualized in R with ggplot2. Scripts for replicating this analysis can be found at https://github.com/cb-scott/PoritesHolobiont_Final and archived on Zenodo (https://doi.org/10.5281/zenodo.15565326).

## Results

### *Porites spp.* host genetic structure predicted by site

Initial population genetic analysis suggested a high degree of clustering. We identified five admixture groups within the individuals we sampled (Figs 2a, 2b, and 3a). Calculation of weighted pairwise $F_{ST}$ further supported genetic distinction

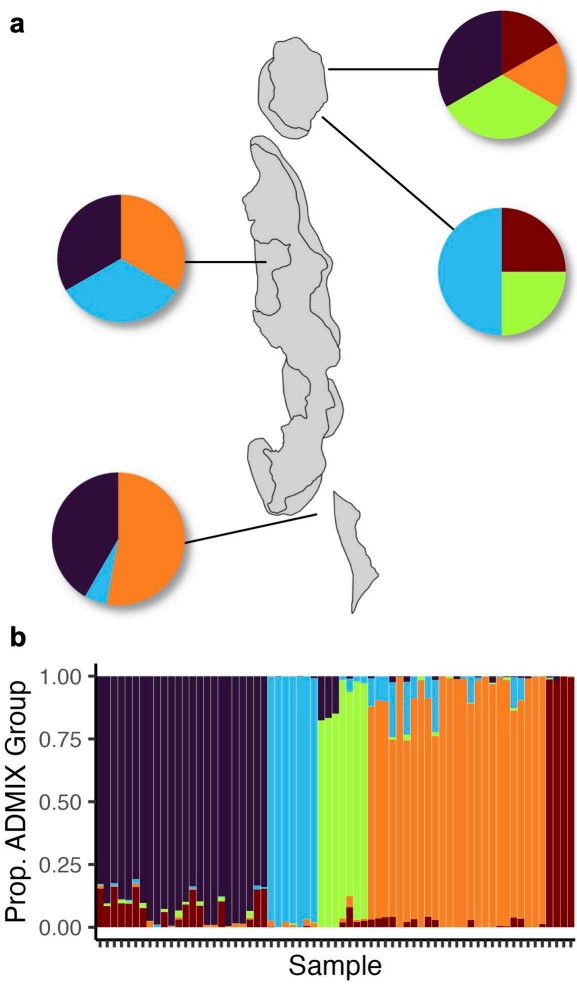

**a**

**b**

Prop. ADMIX Group

Sample

**Fig 2. Host genetics are structured by site.** (a) Proportion of individuals from each admixture group at each site. (b) Admixture plot representing all colonies sequenced, with each vertical bar representing one sample. Bars are ordered by maximum group membership. Orpheus and Pelorus Island map data edited from © Great Barrier Reef Marine Park Authority 2014 [25].

between these five admixture groups (minimum pairwise $F_{ST}$: 0.19, maximum: 0.38, Fig 3 inset, S1 Fig). Through a PER-MANOVA analysis, we determined that site was the only significant (at the $p < 0.05$ level) factor in determining the overall genetic structure of Orpheus and Pelorus Island *Porites* spp. (Figs 2a and 3a, Table 1).

## Holobiont community composition

The vast majority of the Symbiodiniaceae ASVs were identified as C15 (Fig 4). For large *Porites* spp., the Symbiodiniaceae community from colony-level replicates had significantly lower variance than within-admixture or within-site groups (S2 Fig). Microbial communities were dominated by ASVs assigned to *Endozoicomonas* spp. and *Vibrio* spp. (Fig 5). Similar to the Symbiodiniaceae community, within colony variance was significantly lower than within-site or within-admixture variance (S3 Fig).

## Symbiodiniaceae community more structured by host genetics than microbial community

We compared the relative importance of host genetics, size class, and site in structuring the Symbiodiniaceae and microbial community using an RDA-forest model. For both holobiont members, the host's genetic subcluster primarily drove community variation, followed by site, then size class (Fig 6c, S1 and S2 Tables).

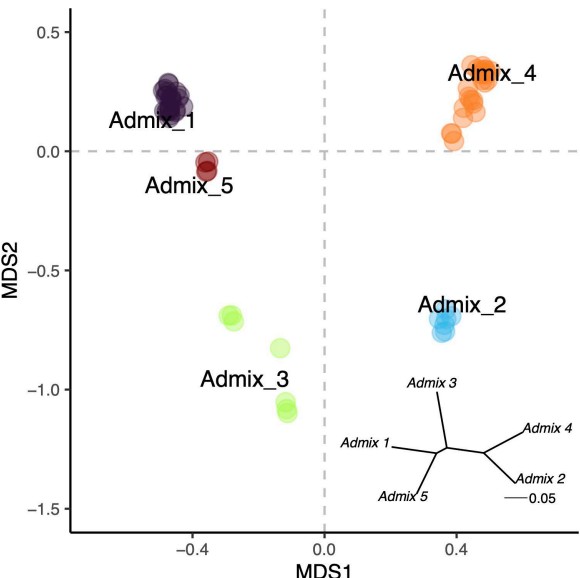

**Fig 3. A large degree of genetic structure exists in *Porites* spp. species complex.** (a) PCoA of genetic distances colored by assigned admixture group. (Inset) Unrooted hierarchical tree of pairwise $F_{ST}$ distances between admixture groups. $F_{ST}$ values are reported in S1 Fig.

**Table 1. Site structures *Porites* spp. host genetics.** PERMANOVA results for the model, Genetic Distance ~ Site + Size Class + Site*Size Class, with the marginal importance of each variable given.

|  | R2 | F | p-value |
| --- | --- | --- | --- |
| Site | 0.076 | 1.76 | 0.006 |
| Size Class | 0.022 | 1.53 | 0.078 |
| Site*Size | 0.021 | 1.43 | 0.107 |
| Residual | 0.881 |  |  |

RDA-forest supported a stronger link between the host's genetic cluster and Symbiodiniaceae community than the microbial community. For the Symbiodiniaceae, of the total gradient forest model importance (total importance = 0.135), 74% of the importance was generated by genetic cluster (weighted variable importance = 0.100), 17% was generated by site (weighted variable importance = 0.023), and 8% was generated by size class (weighted variable importance = 0.011; Table S1). At the same time, for the microbial community, of the total gradient forest model importance (total importance = 0.463), only 52% of the importance was generated by genetic cluster (weighted variable importance = 0.243), 33% was generated by site (weighted variable importance = 0.155), and 14% was generated by size class (weighted variable importance = 0.065; S2 Table). These results were additionally supported by separate PERMANOVA analyses (Tables 2 and 3).

### Differentially abundant microbial taxa between groups

We found 29 significantly differentially abundant bacterial taxa between size classes and/or between sites (S4 Fig). Most notably, some strains of *Vibrio* spp. and *Endozoicomonas* spp. were differentially abundant between sites. Additionally, some strains of *Vibrio* spp. were differentially abundant between size classes.

### Discussion

We found a high degree of genetic structure in massive *Porites*. This is unsurprising, as previous studies have revealed complex genetic structure within the species complex [24,44]. We identified five admixture groups in this study from just

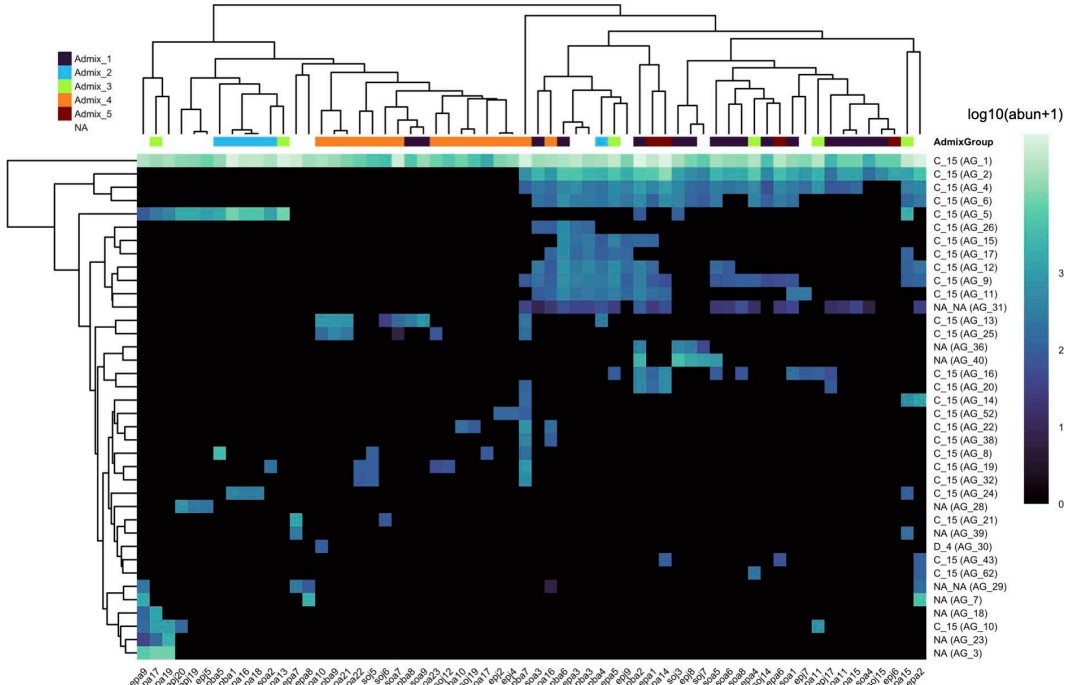

**Fig 4. C15 ASVs dominate Symbiodiniaceae community.** Heatmap of ASV group abundance in each coral colony, ordered by the genetic subcluster each colony was assigned to. Colors along the top of the heatmap indicate which admixture group each colony was assigned to and correspond to the colors in Fig 3. We were not able to recover host genetic data for some colonies which had successful ITS2 amplification. These colonies are marked "NA" and are annotated white. Each row is a different ASV group, labeled by their assigned taxonomy. Most ASV groups belonged to C15 (a *Clado-copium* spp. subtype), though we recovered one ASV group which was assigned to D4 (a *Durusdinium* spp. subtype). Groups labeled "NA" did not have a consensus taxonomy call. Fill colors within the heatmap are log standardized values, $\log_{10}(\text{Abundance}+1)$, for the abundance of each ASV group.

two neighboring islands in the Indo Pacific. Likely, the distinct admixture groups recovered reflect the sampling of multiple species of *Porites* spp., as they are morphologically plastic and notoriously difficult to identify in the field. Some of the genetic clusters that we have identified may correspond to different nominal species of Porites, while others may correspond to cryptic lineages within these species. We could not discriminate between these possibilities for the lack of samples and expertise to analyze *Porites* spp. morphology. We therefore chose to talk about "genetic clusters" rather than species or cryptic lineages (which are supposed to belong to the same nominal species). Much like previous studies, these genetic clusters are associated with specific geographical locations, hinting at an evolutionary mechanism linked to diverse environmental conditions.

For both the Symbiodiniaceae and microbial communities, host genetic cluster was the most important factor in determining community structure (Fig 6c). This aligns with recent work showing that cryptic lineages of *Porites lutea* at environmentally distinct sites in Palau harbor diverging holobiont strategies [44]. This was most stark for the vertically transmitted Symbiodiniaceae community, where over 70% of the predictive power in our RDA-forest model came from genetic cluster. Combined with the PERMANOVA results, we did not find support for site-specific or size-specific Symbiodiniaceae communities. This advances evidence for long-term coevolution between *Porites* spp. and their symbiotic algae [45,46]. All coral colonies were dominated by symbiont strain C15, so differences between genetic clusters are being driven by subgeneric partitioning within C15, background strains, or, to a lesser extent, by symbionts of potentially the same genus or other genera that weren't classified (Fig 4). It is possible that hosting different background Symbiodiniaceae communities may advantage the host during stress-induced "symbiont shuffling" [10,45,47]. However, given host

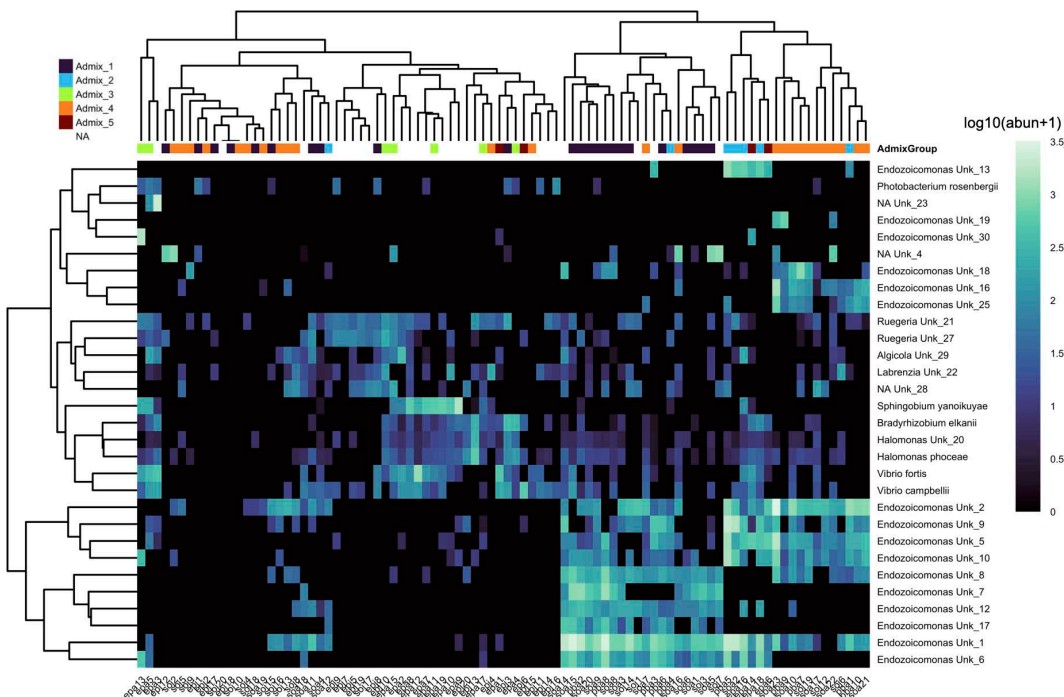

**Fig 5. *Porites* microbial community composition across genetic subclusters.** Heatmap of ASV abundance in each coral colony for the top 30 most prevalent ASVs, ordered by the genetic subcluster each colony was assigned to. Colors along the top of the heatmap indicate which admixture group (top) and genetic subcluster (bottom) each colony was assigned to and correspond to the colors in Fig 3. We were not able to recover host genetic data for some colonies which had successful 16S amplification. These colonies are marked "NA" in the annotation colors. Each row is a different ASV, labeled by its assigned genus- and species-level taxonomy. Groups labeled "NA" or "Unk" did not have a consensus taxonomy call but still represent putatively separate species. Fill colors within the heatmap are log standardized values, $\log_{10}(\text{Abundance} + 1)$, for the abundance of each ASV.

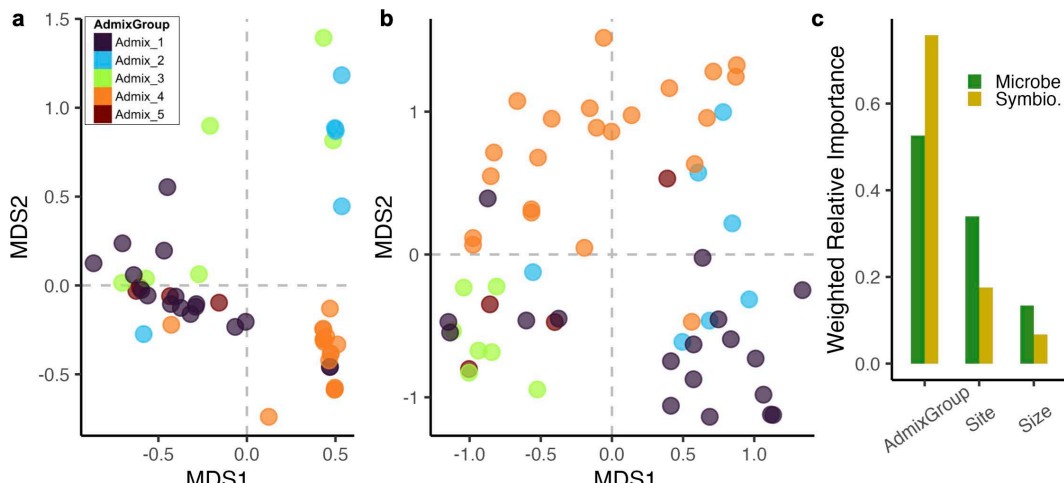

**Fig 6. Host genetics is the primary driver of Symbiodiniaceae and microbial community composition.** (a) PCoA of Symbiodiniaceae community structure, based on Bray-Curtis distances, colored by host colony admixture group. (b) PcoA of microbial community structure, based on Bray-Curtis distances, colored by host colony admixture group. (c) RDA-forest model results giving weighted relative importance for Symbiodiniaceae (gold) and microbial (green) communities. In brief, weighted relative importance represents the proportion of variance explained by a given variable out of the total variance explained by the model. For both microbes and Symbiodiniaceae, genetic subcluster is the most important predictor of community structure. We find qualitatively similar results through PERMANOVA analysis (Tables 2 and 3).

**Table 2. PERMANOVA results for drivers of Symbiodiniaceae community structure support admixture group as most important predictor.** Host genetics have the strongest (and only significant) effect on symbiont community, followed by sampling site, then size class.

| Predictor | $R^2$ | F | p |
|---|---|---|---|
| Admixture Group | 0.394 | 7.42 | 0.001 |
| Site | 0.059 | 1.48 | 0.146 |
| Size Class | 0.015 | 1.16 | 0.288 |
| Residual | 0.531 | | |

**Table 3. PERMANOVA results for drivers of microbial community structure show a significant role of admixture group, site, and size class.** Host genetics have the strongest effect on the microbial community, followed by sampling site, then size class.

| Predictor | $R^2$ | F | p |
|---|---|---|---|
| Admixture Group | 0.132 | 2.56 | 0.001 |
| Site | 0.081 | 2.08 | 0.001 |
| Size class | 0.042 | 3.21 | 0.001 |
| Residual | 0.745 | | |

genetic structure is associated with site (Fig 3a, Table 1), this pattern may simply reflect the environmental abundance of Symbiodiniaceae.

Previous work has identified site-specific, age-dependent microbial diversity patterns in *Porites lutea* [48]. We build on this work, finding a significant effect of site and size class in determining the microbial community structure of massive *Porites* (Fig 6c, Table 3). However, we note that while Wainwright et al. [48] used corallite structure to assign all samples to *P. lutea*, our results suggest there is likely additional genetic substructure within the named *Porites* species. We find that genetic clusters can be structured by site (Table 1), thus these site-specific patterns may partially reflect genetic associations with specific microbial communities. Likely, as we show here, site-specific factors are trumped by host genetics (Fig 6c).

Similarly, the size class differences we found in the microbiome are small, but significant, compared to the role of host genetics. Size class specific communities may reflect changes in the microbial community with host age, and/or morphology-driven changes. These differences may be driven by neutral (to the host) ecological processes, such as priority effects [49], host selection for a specific microbiome over time [8], or colony geometry [50]. Our study highlights that studies of coral holobiont dynamics must account for host genetic background, as fine-scale differences may be obscured by lineage-specific communities. However, our study did not confirm whether small *Porites* on the reef are the direct descendants of concurrent massive individuals. Future studies evaluating the role age and environmental drivers in structuring the holobiont community would be most robust if relatedness between individuals was controlled for (e.g., through the out planting of offspring with known parents).

We found some microbial community differences to be driven by known functionally important taxa (S4 Fig). One such case is the differential abundance of *Vibrio* species between large and small colonies. *Vibrio* spp. is an opportunistic pathogen [51] and has been associated with coral bleaching [52]. This suggests that particular *Vibrio* strains may be age-class specific pathogens, or, more likely to invade age-specific communities. Another case is between-site differences in *Endozoicomonas* spp. abundance. Work quantifying *Endozoicomonas* at the bacterial lineage level has shown that *Endozoicomonas* strains are often geographically distinct in *Porites* corals, and some strains may have genes indicating their involvement in the sulfur cycle [53]. This suggests that the between-site differences we recovered may reflect functional, strain-level differences related to the local environment. However, we were only able to classify *Endozoicomonas* ASVs to the genus level with our data and massive diversity exists within the genus. It is clear that *Endozoicomonas* species can

span the mutualism to parasitism continuum [54]. While some species of *Endozoicomonas* may be beneficial microbial symbionts modulating the environment for their coral hosts, others may be parasites of marine taxa or opportunistic colonizers from the environment [5,55,56]. To make stronger claims about the role of *Endozoicomonas* to the *Porites* holobiont in different environments, future work should generate metagenomic datasets to assess species and functional diversity while controlling for host genetic background.

Above all, our results stress that failure to assess genetic structure may lead to the overestimation of holobiont flexibility. Cryptic species complexes are emerging as a common feature of scleractinian taxa and often segregate by site/environment [24,57–59]. Thus, site-specific patterns in the holobiont may not reflect flexible environmental associations, but rather genetically determined, lineage-specific communities. It is imperative to genotype hosts in holobiont studies, as morphologically similar – but genetically divergent – sympatric corals are likely associated with unique microbial/microalgal communities.

## Conclusion

We demonstrated that host genetic structure is the most important factor in determining both coral-associated Symbiodiniaceae and microbial communities in massive *Porites* at Orpheus and Pelorus Islands, Australia. Local environment and size of the host have much less effect, although the microbial community is more responsive to local environment than the association with Symbiodiniaceae. This strong fidelity to host genetic clusters suggests the holobiont components cannot easily evolve independently from the host. Thus, underlying host genetic structure must always be weighed in determining the acclimatization potential of the holobiont to future conditions.

## Supporting information

**S1 File.  Supplemental Figures and Tables.** All supporting figures and tables referenced throughout the text are located in this file.
(PDF)

## Acknowledgments

The data analysis for this manuscript has been performed using facilities of the Texas Advanced Computing Center (TACC). We thank Kristina Black, Greg Torda, and JP Rippe for their help collecting these samples and their camaraderie in the field and the staff of Orpheus Island Research Station for supporting the logistics of this project.

## Author contributions

**Conceptualization:** Carly B Scott, Mikhail V Matz.

**Data curation:** Carly B Scott, Raegen Schott.

**Formal analysis:** Carly B Scott, Raegen Schott.

**Funding acquisition:** Carly B Scott, Mikhail V Matz.

**Investigation:** Carly B Scott, Raegen Schott.

**Methodology:** Mikhail V Matz.

**Project administration:** Mikhail V Matz.

**Resources:** Mikhail V Matz.

**Software:** Carly B Scott, Raegen Schott, Mikhail V Matz.

**Supervision:** Mikhail V Matz.

**Validation:** Mikhail V Matz.

**Visualization:** Carly B Scott.

**Writing – original draft:** Carly B Scott.

**Writing – review & editing:** Carly B Scott, Mikhail V Matz.

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
