## [Decision Letter · Decision Letter 0]

PONE-D-24-28640Cryptic genetic structure of the coral host is the primary driver of holobiont assembly in massive PoritesPLOS ONE

Dear Dr. Scott,

Thank you for submitting your manuscript to PLOS ONE. We invite you to submit a revised version of the manuscript that addresses the (minor) points raised during the review process.

We look forward to receiving your revised manuscript.

Kind regards,

Frank Melzner

Academic Editor

PLOS ONE

Journal Requirements:

This research was supported by NSF DGE-2137420 to C.B.S, NSF grant IOS-1755277 to M. V. M., funding from the University of Texas Department of Integrative Biology to C.B.S, and funding from the International Women’s Fishing Association to C.B.S. 

This research was supported by NSF grant DGE-2137420 to C.B.S, NSF grant IOS-1755277 to M. V. M., funding from the University of Texas Department of Integrative Biology to C.B.S, and funding from the International Women’s Fishing Association to C.B.S. The data analysis has been performed using facilities of the Texas Advanced Computing Center (TACC). We thank Kristina Black, Greg Torda, and JP Rippe for their help collecting these samples and their camaraderie in the field, and, the staff of Orpheus Island Research Station for supporting the logistics of this project. 

This research was supported by NSF DGE-2137420 to C.B.S, NSF grant IOS-1755277 to M. V. M., funding from the University of Texas Department of Integrative Biology to C.B.S, and funding from the International Women’s Fishing Association to C.B.S.

5. In the online submission form, you indicated that your data will be submitted to a repository upon acceptance.  We strongly recommend all authors deposit their data before acceptance, as the process can be lengthy and hold up publication timelines. Please note that, though access restrictions are acceptable now, your entire minimal  dataset will need to be made freely accessible if your manuscript is accepted for publication. This policy applies to all data except where public deposition would breach compliance with the protocol approved by your research ethics board. If you are unable to adhere to our open data policy, please kindly revise your statement to explain your reasoning and we will seek the editor's input on an exemption. 

6. We note that Figures 1, 2 in your submission contain [map/satellite] images which may be copyrighted. All PLOS content is published under the Creative Commons Attribution License (CC BY 4.0), which means that the manuscript, images, and Supporting Information files will be freely available online, and any third party is permitted to access, download, copy, distribute, and use these materials in any way, even commercially, with proper attribution. For these reasons, we cannot publish previously copyrighted maps or satellite images created using proprietary data, such as Google software (Google Maps, Street View, and Earth). For more information, see our copyright guidelines: http://journals.plos.org/plosone/s/licenses-and-copyright.

a. You may seek permission from the original copyright holder of Figures 1, 2 to publish the content specifically under the CC BY 4.0 license.  

Reviewers' comments:

Reviewer's Responses to Questions

**Comments to the Author**

1. Is the manuscript technically sound, and do the data support the conclusions?

Reviewer #1: Yes

Reviewer #2: Partly

2. Has the statistical analysis been performed appropriately and rigorously? 

Reviewer #1: Yes

Reviewer #2: Yes

3. Have the authors made all data underlying the findings in their manuscript fully available?

Reviewer #1: Yes

Reviewer #2: Yes

4. Is the manuscript presented in an intelligible fashion and written in standard English?

Reviewer #1: Yes

Reviewer #2: Yes

5. Review Comments to the Author

Reviewer #1: The presented study provides an overview of cryptic lineages in massive Porites spp. on Orpheus and Pelorus Islands on the Great Barrier Reef, Australia. Cryptic diversity is large in corals in general, but specifically in massive Porites as other studies have found. The paper presented here expands on these previous studies and shows that the genetic structure of the host was the main driver of symbiont and microbial associations. They further disentangle the drivers reef site and size class (small, i.e. <30cm and large, i.e. >2m) on the associated microbes, where the microbiome was shaped by reef site and size class and the symbiont community was unaffected by both. In light of growing restauration efforts and other techniques to mitigate the impact of climate change on coral reefs, this study adds onto the relevant literature identifying cryptic diversity in corals and provides a framework for understanding holobiont flexibility in light of environmental change. The paper is well-written, the methodology is thorough and the conclusions drawn are comprehensible. I would advise to accept this manuscript with minor changes.

Minor issues are provided in the attached file.

Reviewer #2: The ms presented by Scott et al. is a well written and data rich study that investigates factors responsible for prokaryotic and algal microbiome composition of Porites spp. corals. There has been relatively little focus to dissect detailed microbial and algal strain diversity in relationship to fine scale host genetics. In that respect, the study is timely and interesting to the coral community and beyond. The methodology looks sound to me, the presentations are generally clear.

My main concern is the notion of “cryptic genetic structure” that already appears in the title. My understanding is that the authors study a species complex, genus Porites, where the existence of several species is an acknowledged fact. Being not a coral expert, I assume that these species are morphologically so plastic and /or similar that non-genetic taxonomy will not work. This is also rightly acknowledged by the authors in line 366.

By means of Rad-seq /SNP calling they are able to resolve them to species level which is an interesting finding in and among itself.

If so, it is rather (the difficult to determine) taxonomic affiliation to a species within a genus that determines the composition of algal and prokaryotic symbionts. As such, the entire data set would also be interesting data worthwhile to report, but with a changed interpretation /implication. As it stands now, the title implies (to me) within-species genetic structure.

Along the same lines, in order to benchmark the differences among the host genetic clusters (aka cryptic species?) it would be great to quantify /date the divergence among coral hosts, maybe with some mitochondrial genetic clock? This way, the data would become comparable to other such studies that look at the effects of host genotypic /species affiliation for microbial community assembly.

Finally, can any specific inference be made as to the level of species-affiliated genomic divergence vs. within species genetic structure, based on the data of the host? I note that there are two levels of clustering that the authors describe, but this is not picked up later in the ms. Maybe the combined sample size is too low here?

Further suggestions, in the order of appearance

L58: the study results do not really present anything on the role of the associated microbial community for holobiont adaptation /resilience, so a different intro lead would be more appropriate for the paper

L88: This reads awkward, as the data presented are also not long term, but deal with a single survey

L99: an overview on the “site * colony size * replicate sample” combinations would be nice, and also the absolute spatial distances at all scales.

Fig 4. Here, the community composition patterns look like rather uniform among host groups. This might partly be a result of the “binning” of the color coding. But this is confusing as the diagram in its present form suggests hardly any structure. This figure should convey as much qualitative differentiation as possible to explain the clear MDS clustering in later analyses.

As a suggestion, is it possible to look at a heat map for each individual sample clustered (i) by microbial /algal composition (ii) by host genetics? Such a diagram could convey everyting at a glance.

L363: I would present a STRUCTURE graph also with 10 clusters, as you repeatedly refer to that optimal cluster number rather than 5 (as in Fig 2).

6. PLOS authors have the option to publish the peer review history of their article (what does this mean? ). If published, this will include your full peer review and any attached files.

**Do you want your identity to be public for this peer review?** For information about this choice, including consent withdrawal, please see our Privacy Policy .

Reviewer #1: No

Reviewer #2: No

---

## [Author Response · Author response to Decision Letter 1]

16 Jun 2025

To the Editor(s) at PLOS ONE:

Thank you for securing detailed reviews for our manuscript, “Cryptic genetic structure of the coral host is the primary driver of holobiont assembly in massive Porites,” for publication in PLOS ONE. We appreciate the insightful comments left by the reviewers, which have improved the clarity and interpretation of our results. We have incorporated most of the suggestions provided by the reviewers, which are highlighted within the manuscript. Please see below, in blue, for a point-by-point response to the reviewers’ comments. We note that Reviewer 1 suggested many small grammatical corrections which have improved the readability of our manuscript. Rather than respond to each of these suggestions individually below, we have simply made all the requested grammatical changes in the manuscript. We additionally note that while all content changes have been noted in the tracked changes version of the manuscript, we further edited style only in the clean version of the manuscript submitted to better reflect PLOS ONE’s style guidelines.

Additionally, in response to editor comments from PLOS ONE:

1. We have updated our funding disclosure statement and edited our acknowledgements to remove funding information. This is reflected in the cover page to our paper. The new funding statement now reads:

This research was supported by NSF DGE 2137420 to C.B.S, NSF grant IOS-1755277 to M. V. M., funding from the University of Texas Department of Integrative Biology to C.B.S, and funding from the International Women’s Fishing Association to C.B.S. The funders had no role in the study design, data collection and analysis, decision to publish, or preparation of the manuscript.

The new acknowledgements statement reads:

The data analysis for this manuscript has been performed using facilities of the Texas Advanced Computing Center (TACC). We thank Kristina Black, Greg Torda, and JP Rippe for their help collecting these samples and their camaraderie in the field and the staff of Orpheus Island Research Station for supporting the logistics of this project.

2. We have published our data in open access repositories. All sequencing data can now be found at PRJNA1048506 and processing scripts are on GitHub at https://github.com/cb-scott/PoritesHolobiont_Final and on Zenodo at 10.5281/zenodo.15565326. The data availability statement has been updated to reflect this.

3. Figures 1 & 2 contain satellite image data. We created these maps ourselves from raw geospatial files provided by the Great Barrier Marine Park Authority (https://catalogue.eatlas.org.au/geonetwork/srv/eng/catalog.search#/metadata/ac8e8e4f-fc0e-4a01-9c3d-f27e4a8fac3c). This data is protected by a Creative Commons – Attribution 4.0 International license, which if we understand correctly, enables the redistribution of the material in any medium or format for any purpose. Please let us know if this is not the case. To clarify this, we have included a reference to this raw dataset in our methods and figure captions. The map of Australia in Figure 1 was sourced from the ‘maps’ package data in R. This data comes from the Natural Earth Data project (https://www.naturalearthdata.com/), which is a public domain map dataset.

Reviewer 1

Reviewer #1: The presented study provides an overview of cryptic lineages in massive Porites spp. on Orpheus and Pelorus Islands on the Great Barrier Reef, Australia. Cryptic diversity is large in corals in general, but specifically in massive Porites as other studies have found. The paper presented here expands on these previous studies and shows that the genetic structure of the host was the main driver of symbiont and microbial associations. They further disentangle the drivers reef site and size class (small, i.e. <30cm and large, i.e. >2m) on the associated microbes, where the microbiome was shaped by reef site and size class and the symbiont community was unaffected by both. In light of growing restauration efforts and other techniques to mitigate the impact of climate change on coral reefs, this study adds onto the relevant literature identifying cryptic diversity in corals and provides a framework for understanding holobiont flexibility in light of environmental change. The paper is well-written, the methodology is thorough, and the conclusions drawn are comprehensible. I would advise to accept this manuscript with minor changes.

General comments:

You should use a grammar checking tool. I have underlined some of the issues in the following, but not all. The issues are minor, but sometimes you are missing a “the” or an “of”. Further, make sure to decide on either “2bRAD” or “2b-RAD” and use it throughout. And, often there is a space between “>” and the following value, but sometimes there isn’t. Decide on one and keep the scheme throughout. The same goes for units, please add a space between the value and its corresponding unit in all instances. Lastly, please check that Porites is in italic in all instances. I added some below but did not comment on each case.

The reviewer’s attention to detail is appreciated, we have made the requested changes throughout and checked for consistency. The reviewer also suggested many minor grammatical and phrasing changes in their comments. We have made all of the suggested changes throughout the manuscript (and have highlighted them in a tracked-changes version) but have not responded to each individual grammatical comment in their review.

Introduction

General comment: You make the case that long-term studies are lacking to disentangle shifts in the microbial communities of coral holobionts, but it is difficult to make your case without confirming whether the juveniles are direct offspring of the large colonies. In terms of symbionts it is possible, since you show stable associations with specific symbiont types and this is generally the case in many corals, but the microbiome is a different story. So I would suggest to add a couple of sentences either here or in the discussion about the limitations of such inferences.

We have added some text to the discussion about the limitation of such inferences and some recommendations for future studies on lines 318- 323 of the revised manuscript:

Our study highlights that studies of coral holobiont dynamics must account for host genetic background, as fine-scale differences may be obscured by lineage-specific communities. However, our study did not confirm whether small Porites on the reef are the direct descendants of concurrent massive individuals. Future studies evaluating the role age and environmental drivers in structuring the holobiont community would be most robust if relatedness between individuals was controlled for (e.g., through the out planting of offspring with known parents).

Methods

ll. 159ff: I would give a reasoning for the genomes you used. E.g. why did you choose P. lutea as a reference? If massive Porites in Australia have not been identified to species level would it have been beneficial to combine multiple Porites reference genomes into your reference? I mainly ask since P. lobata is generally used for Porites sp. and the reference genome published by Noel et al. (2023) has a higher completeness and continuity than P. lutea.

We appreciate that the reviewer has highlighted a more complete reference. Quite simply, we did not use this reference as it was not yet published when we began our analysis. We remapped our data to the updated reference and evaluated initial population genetic structure. This yielded qualitatively the same placement of samples in PCA space. This figure is included in the 'Response to Reviewers' letter uploaded.

Given the similarity of sample placement, we retained our analysis using the original genome.

Discussion

LL. 378-379: I would add here that the differences between genetic clusters is driven by subgeneric partitioning within C15 and to a lesser extent by symbionts of potentially the same genus or other genera that weren’t classified.

The reviewer is correct in their interpretation of our results – the patterns we observed could have been due to partitioning within C15, by the differences in background symbiont communities, or by differences in unclassified taxa. We have changed our discussion text to read on lines 300 – 302 of the revised manuscript:

All coral colonies were dominated by symbiont strain C15, so differences between genetic subclusters are being driven by subgeneric partitioning within C15, background strains, or, to a lesser extent, by symbionts of potentially the same genus or other genera that weren’t classified (Figure 4).

LL. 406ff: I would expand a bit more on the differences in Endozoicomonas abundance between the sites. Since there is quite some literature on Endozoicomonas now, it should be mentioned briefly here.

The reviewer’s comments on the body of literature on Endozoicomonas are well merited. We had initially hesitated to make too strong of claims about Endozoicomonas function as we have only classified diversity to the genus level and there is significant debate about the role of specific Endozoicomonas species. We have included a more in-depth discussion of this evolving literature and possible limitations on lines 328—339 of the revised manuscript, in lieu of the original text on lines 406 - 410:

Work quantifying Endozoicomonas at the bacterial lineage level has shown that Endozoicomonas strains are often geographically distinct in Porites corals, and some strains may have genes indicating their involvement in the sulfur cycle (Hochart et al., 2023). This suggests that the between-site differences we recovered may reflect functional, strain-level differences related to the local environment. However, we were only able to classify Endozoicomonas OTUs to the genus level with our data and massive diversity exists within the genus. It is clear that Endozoicomonas species can span the mutualism to parasitism continuum (Pogoreutz & Ziegler, 2024). While some species of Endozoicomonas may be beneficial microbial symbionts modulating the environment for their coral hosts, others may be parasites of marine taxa or opportunistic colonizers from the environment (Peixoto et al., 2017; Tandon et al., 2022; Katharios et al., 2015). To make stronger claims about the role of Endozoicomonas to the Porites holobiont in different environments, future work should generate metagenomic datasets to assess species and functional diversity while controlling for host genetic background.

Figures

Fig. 4: Please edit the spelling within the figure to “ASVs” instead of “ASVS”. Further, for comprehension I would add the description of D4 ASVs to the figure legend.

In response to Reviewer 2, we have included an updated version of this figure which also corrects this error.

Fig. 6: You show the admixture group colours in Fig. 3 but I would add them here too. I think it would benefit the figure to be able to understand it without the help of another figure.

We have added a legend to this figure, corresponding to the admixture group colors defined in Figures 3, 4, and 5.

Reviewer 2

Reviewer #2: The ms presented by Scott et al. is a well written and data rich study that investigates factors responsible for prokaryotic and algal microbiome composition of Porites spp. corals. There has been relatively little focus to dissect detailed microbial and algal strain diversity in relationship to fine scale host genetics. In that respect, the study is timely and interesting to the coral community and beyond. The methodology looks sound to me, the presentations are generally clear.

My main concern is the notion of “cryptic genetic structure” that already appears in the title. My understanding is that the authors study a species complex, genus Porites, where the existence of several species is an acknowledged fact. Being not a coral expert, I assume that these species are morphologically so plastic and /or similar that non-genetic taxonomy will not work. This is also rightly acknowledged by the authors in line 366.

By means of Rad-seq /SNP calling they are able to resolve them to species level which is an interesting finding in and among itself. If so, it is rather (the difficult to determine) taxonomic affiliation to a species within a genus that determines the composition of algal and prokaryotic symbionts. As such, the entire data set would also be interesting data worthwhile to report, but with a changed interpretation /implication. As it stands now, the title implies (to me) within-species genetic structure.

The reviewer is correct that we cannot formally distinguish between nominal (very hard to identify) Porites species and cryptic lineages within each species. We agree that it is confusing that we call our genetic clusters “cryptic lineages”, and that some wording needs to be changed.

To correct this confusion, we have changed “cryptic lineages” to “genetic clusters” throughout and have added the following text into the Discussion to highlight this:

Lines 284-292: Likely, the distinct admixture groups recovered reflect the sampling of multiple species of Porites spp., as they are morphologically plastic and notoriously difficult to identify in the field. Some of the genetic clusters that we have identified may correspond to different nominal species of Porites, while others may correspond to cryptic lineages within these species. We could not discriminate between these possibilities for the lack of samples and expertise to analyze Porites spp. morphology. We therefore chose to talk about “genetic clusters” rather than species or cryptic lineages (which are supposed to belong to the same nominal species).

We have additionally changed the wording of the title to reflect this shift in interpretation:

Genetic clustering within massive Porites species complex is the primary driver of holobiont assembly

Along the same lines, in order to benchmark the differences among the host genetic clusters (aka cryptic species?) it would be great to quantify /date the divergence among coral hosts, maybe with some mitochondrial genetic clock? This way, the data would become comparable to other such studies that look at the effects of host genotypic /species affiliation for microbial community assembly.

We agree with the reviewer that quantifying the divergence among coral hosts with a mitochondrial clock would increase this study’s comparability with existing work. However, with our existing 2bRAD data, we only recovered one overlapping site between samples which mapped to the Porites lutea mitochondrial genome, which was insufficient for downstream analysis. In an attempt to quantify divergence, we calculated pairwise FST between genetic clusters. These calculations are included in Supplemental Figure 1. Additionally, we have visualized the pairwise FST values between genetic clusters/admixture groups as an unrooted tree in panel ‘b’ of Figure 3.

Finally, can any specific inference be made as to the level of species-affiliated genomic divergence vs. within species genetic structure, based on the data of the host? I note that there are two levels of clustering that the authors describe, but this is not picked up later in the ms. Maybe the combined sample size is too low here?

We appreciate the reviewer’s concerns about the two levels of genetic clustering. Upon revisiting this topic (combined with the feedback below on the admixture plot), we have decided to temper our claim about genetic clustering within the admixture groups. This comment prompted us to reevaluate our clustering methodology, and we realized that clustering outcomes beyond the five admixture clusters are sensitive to methods and parameters chosen. Therefore, we decided to fall back to the five admixture clusters to represent the genetic structure in our data. We had already run this analysis in parallel with the original analysis that we presented and had recovered qualitatively the same result (in the supplemental figures and tables). We have modified the manuscript to instead center the admixture group-based analysis in the main text and figures, as we feel like the determination of the five primary genetic clusters is less subjective. We reran all analyses and remade all the figures. The r

---

## [Editor Report · Decision Letter 1]

Genetic clustering within massive Porites species complex is the primary driver of holobiont assembly

PONE-D-24-28640R1

Dear Dr. Scott,

We’re pleased to inform you that your manuscript has been judged scientifically suitable for publication and will be formally accepted for publication once it meets all outstanding technical requirements.

Kind regards,

Frank Melzner

Academic Editor

PLOS ONE
---

## [Editor Report · Acceptance letter]

PONE-D-24-28640R1

PLOS ONE

Dear Dr. Scott,

I'm pleased to inform you that your manuscript has been deemed suitable for publication in PLOS ONE. Congratulations! Your manuscript is now being handed over to our production team.

Kind regards,

on behalf of

Dr. Frank Melzner

Academic Editor

PLOS ONE